# CONVOLUTIONAL CRFS FOR SEMANTIC SEGMENTATION

## ABSTRACT

For the challenging semantic image segmentation task the best performing models have traditionally combined the structured modelling capabilities of Conditional Random Fields (CRFs) with the feature extraction power of CNNs. In more recent works however, CRF post-processing has fallen out of favour. We argue that this is mainly due to the slow training and inference speeds of CRFs, as well as the difficulty of learning the internal CRF parameters. To overcome both issues we propose to add the assumption of conditional independence to the framework of fully-connected CRFs. This allows us to reformulate the inference in terms of convolutions, which can be implemented highly efficiently on GPUs .Doing so speeds up inference and training by two orders of magnitude. All parameters of the convolutional CRFs can easily be optimized using backpropagation. Towards the goal of facilitating further CRF research we have made our implementations publicly available.

## 1 INTRODUCTION

Semantic image segmentation, which aims to produce a categorical label for each pixel in an image, is a very import task for visual perception. Convolutional Neural Networks have been proven to be very strong in tackling semantic segmentation tasks (Long et al., 2015; Chen et al., 2018; 2017; Zhao et al., 2017). While simple feed-forward CNNs are extremely powerful in extracting local features and performing good predictions utilizing a small field of view, they lack the capability to utilize context information and cannot model interactions between predictions directly. Thus it has been suggested that such deep neural networks may not be the perfect model for structured predictions tasks such as semantic segmentation (Zhao et al., 2017; Lin et al., 2016; Zheng et al., 2015). Several authors have successfully combined the effectiveness of CNNs to extract powerful features, with the modelling power of CRFs in order to address the discussed issues (Lin et al., 2016; Chandra & Kokkinos, 2016; Zheng et al., 2015). Despite their indisputable success, structured models have fallen out of favour in more recent approaches (Wu et al., 2016; Chen et al., 2017; Zhao et al., 2017).

We believe that the main reasons for this development are that CRFs are notoriously slow and hard to optimize. Learning the features for the structured component of the CRF is an open research problem (Vemulapalli et al., 2016; Lin et al., 2016) and many approaches rely on entirely hand-crafted Gaussian features (Krähenbühl & Koltun, 2011; Zheng et al., 2015; Schwing & Urtasun, 2015; Chen et al., 2018). In addition, CRF inference is typically two orders of magnitude slower than CNN inference. This makes CRF based approaches too slow for many practical applications. The long training times of the current generation of CRFs also make more in-depth research and experiments with such structured models impractical.

To solve both of these issues we propose to add the strong and valid assumption of conditional independence to the existing framework of fully-connected CRFs (FullCRFs) introduced by Krähenbühl & Koltun (2011). This allows us to reformulate a large proportion of the inference as convolutions, which can be implemented highly efficiently on GPUs. We call our method convolutional CRFs (ConvCRFs). Backpropagation (Rumelhart et al., 1986) can be used to train all parameters of the ConvCRF. Inference in ConvCRFs can be performed in less then 10ms. This is a speed increase of two-orders of magnitude compared to FullCRFs. We believe that those fast train and inference speeds will greatly benefit future research and hope that our results help to revive CRFs as a popular method to solve structured tasks.

## 2 RELATED WORK

Recent advances in semantic segmentation are mainly driven by powerful deep neural network architectures (Krizhevsky et al., 2012; Simonyan & Zisserman, 2015; He et al., 2016; Wu et al., 2016). Following the ideas introduced by Long et al. (2015), transposed convolution layers are applied at the end of the prediction pipeline to produce high-resolution output. Atrous (dilated) convolutions (Chen et al., 2015a; Yu & Koltun, 2015) are commonly applied to preserve spatial information in feature space.

Many architectures have been proposed (Noh et al., 2015; Ronneberger et al., 2015; Badrinarayanan et al., 2017; Paszke et al., 2016; Teichmann et al., 2016; Wu et al., 2016), based on the ideas above. All of those approaches have in common that they primarily rely on the powerful feature extraction provided by CNNs. Predictions are pixel-wise and conditionally independent (given the common feature base of nearby pixels). Structured knowledge and background context is ignored in these models.

One popular way to integrate structured predictions into CNN pipelines is to apply a fully-connected CRF (FullCRF) (Krähenbühl & Koltun, 2011) on top of the CNN prediction (Chen et al., 2015a; Zheng et al., 2015; Schwing & Urtasun, 2015; Lin et al., 2016; Chandra & Kokkinos, 2016). Utilizing the edge-awareness of CRFs, FullCRFs have been successfully utilized to solve weakly- and semi-supervised segmentation tasks (Li et al., 2018; Triggs & Verbeek, 2008; He & Zemel, 2009; Tang et al., 2018). Tang et al. (2018) propose to use a CRF based loss function. All of those approaches can benefit from our contributions.

**Parameter Learning in CRFs**  FullCRFs rely on hand-crafted features for the pairwise (Gaussian) kernels. In their first publication Krähenbühl & Koltun (2011) optimized the remaining parameters with a combination of expectation maximization and grid-search. In a follow-up work Krähenbühl & Koltun (2013) proposed to use gradient descent. The idea utilizes, that for the message passing the equation $(k_G * Q)' = k_G * Q'$ holds. This allows them to train all internal CRF parameters, using backpropagation, without being required to compute gradients with respect to the Gaussian kernel $k_G$. However the features of the Gaussian kernel cannot be learned with such an approach. CRFasRNN (Zheng et al., 2015) uses the same ideas to implement joint CRF and CNN training. Like Krähenbühl and Koltuns (2013) approach this requires hand-crafted pairwise (Gaussian) features.

Quadratic optimization (Chandra & Kokkinos, 2016; Vemulapalli et al., 2016) has been proposed to learn the Gaussian features of FullCRFs. These approaches however do not fit well into many deep learning pipelines. Another way of learning the pairwise features is piecewise training (Lin et al., 2016). An additional advantage of this method is that it avoids repeated CRF inference, speeding up the training considerably. This approach is however of an approximate nature and inference speed is still very slow.

**Inference speed of CRFs**  In order to circumvent the issue of very long training and inference times, some CRF based pipelines produce an output which is down-sampled by a factor of $8 \times 8$ (Chandra & Kokkinos, 2016; Lin et al., 2016). This speeds up the inference considerably. However this harms their predictive capabilities. Deep learning based semantic segmentation pipelines perform best when they are challenged to produce a full-resolution prediction (Long et al., 2015; Yu & Koltun, 2015; Chen et al., 2017). To the best of our knowledge, no significant progress in inference speed has been made since the introduction of FullCRFs (Krähenbühl & Koltun, 2011).

## 3 FULLY CONNECTED CRFS

In the context of semantic segmentation most recent CRF based approaches are based on the Fully Connected CRF (FullCRF) model introduced by Krähenbühl & Koltun (2011). Consider an input image $I$ consisting of $n$ pixels and a segmentation task with $k$ classes. A segmentation of $I$ is then modelled as a random field $\mathbf{X} = \{X_1, \ldots, X_n\}$, where each random variable $X_i$ takes values in $\{1, \ldots, k\}$, i.e. the label of pixel $i$. Solving $\mathrm{argmax}_X P(X|I)$ then leads to a segmentation $X$ of the input image $I$. $P(X|I)$ is modelled as a CRF over the Gibbs distribution:

$$P(X = \hat{x} | \tilde{I} = I) = \frac{1}{Z(I)} exp(-E(\hat{x}|I)) \tag{1}$$

where the energy function $E(\hat{x}|I)$ is given by

$$E(\hat{x}|I) = \sum_{i \leq n} \psi_u(\hat{x}_i|I) + \sum_{i \neq j \leq N} \psi_p(\hat{x}_i, \hat{x}_j|I). \tag{2}$$

The function $\psi_u(x_i|I)$ is called unary potential. The unary itself can be considered a segmentation of the image and any segmentation pipeline can be used to predict the unary. In practise most newer approaches (Chen et al., 2018; Schwing & Urtasun, 2015; Zheng et al., 2015) utilize CNNs to compute the unary.

The function $\psi_p(x_i, x_j|I)$ is the pairwise potential. It accounts for the joint distribution of pixels $i, j$. It allows us to explicitly model interactions between pixels, such as pixels with similar colour are likely the same class. In FullCRFs $\psi_p$ is defined as weighted sum of Gaussian kernels $\mathbf{k}_g^{(1)} \ldots \mathbf{k}_g^{(m)}$:

$$\psi_p(x_i, x_j|I) := \mu(x_i, x_j) \sum_{m=1}^{M} w^{(m)} \mathbf{k}_g^{(m)}(\mathbf{f}_i^I, \mathbf{f}_j^I), \tag{3}$$

where $w^{(m)}$ are learnable parameters. The feature vectors $\mathbf{f}_i^I$ can be chosen arbitrarily and may depend on the input Image $I$. The function $\mu(x_i, x_j)$ is the compatibility transformation, which only depends on the labels $x_i$ and $x_j$, but not on the image $I$.

A very widely used compatibility function (Krähenbühl & Koltun, 2011; Chen et al., 2018; Zheng et al., 2015) is the Potts model $\mu(x_i, x_j) = 1_{[xi \neq xj]}$. This model tries to assign pixels with similar features the same prediction. Zheng et al. (2015) propose to use $1 \times 1$ convolutions as compatibility transformation. Such a function allows the model to learn more structured interactions between predictions.

FullCRFs utilize two Gaussian kernels with hand-crafted features. The appearance kernel uses the raw colour values $I_j$ and $I_i$ as features. The smoothness kernel is based on the spatial coordinates $p_i$ and $p_j$. The entire pairwise potential is then given as:

$$\mathbf{k}(\mathbf{f}_i^I, \mathbf{f}_j^I) := w^{(1)} exp\left(-\frac{|p_i - p_j|^2}{2\theta_\alpha^2} - \frac{|I_i - I_j|^2}{2\theta_\beta^2}\right) + w^{(2)} exp\left(-\frac{|p_i - p_j|^2}{2\theta_\gamma^2}\right), \tag{4}$$

where $w^{(1)}$, $w^{(2)}$, as well as $\theta_\alpha$, $\theta_\beta$ and $\theta_\gamma$ are the only learnable parameters of the model. Most CRF based segmentation approaches (Chen et al., 2018; Zheng et al., 2015; Schwing & Urtasun, 2015) utilize the very same handcrafted pairwise potentials proposed by Krähenbühl & Koltun (2011). CRFs are notoriously hard to optimize and utilizing hand-crafted features circumvents this problem.

### 3.1 MEAN FIELD INFERENCE

Inference in FullCRFs is achieved using the mean field algorithm (see Algorithm 1). All steps of algorithm 1, other then the message passing, are highly parallelized and can be implemented easily and efficiently on GPUs using standard deep learning libraries. (For details see (Zheng et al., 2015)).

The message passing however is the bottleneck of the CRF computation. Exact computation is quadratic in the number of pixels and therefore infeasible. Krähenbühl & Koltun (2011) instead proposed to utilize the permutohedral lattice (Adams et al., 2010) approximation, a high-dimensional filtering algorithm. The permutohedral lattice however is based on a complex data structure. While there is a very sophisticated and fast CPU implementation, the permutohedral lattice does not follow the SIMD (Nickolls et al., 2008) paradigm of efficient GPU computation. In addition, efficient gradient computation of the permutohedral lattice approximation, is also a non-trivial problem. This is the underlying reason why FullCRF based approaches use hand-crafted features.

---

**Algorithm 1** Mean field approximation in fully connected CRFs

---

1: Initialize: $\qquad\qquad\qquad\qquad\qquad\qquad\qquad\qquad\quad \triangleright \tilde{Q}_i \leftarrow \frac{1}{Z_i} exp(-\psi_u(x_i|I))$ "softmax"

2: **while** not converged **do**

3: $\qquad \tilde{Q}_i(l) \leftarrow \sum_{i \neq j} w^{(m)} \mathbf{k}_g^{(m)}(\mathbf{f}_i^I, \mathbf{f}_j^I) \tilde{Q}_i(l) \qquad\qquad\qquad\qquad\qquad \triangleright$ Message Passing

4: $\qquad \tilde{Q}_i(x_i) \leftarrow \sum_{l' \in L} \mu(x_i, l') \tilde{Q}_i(l) \qquad\qquad\qquad\qquad \triangleright$ Compatibility Transformation

5: $\qquad \tilde{Q}_i(x_i) \leftarrow \psi_u(x_i|I) + \tilde{Q}_i(x_i) \qquad\qquad\qquad\qquad\quad \triangleright$ Adding Unary Potentials

6: $\qquad \tilde{Q}_i(x_i) \leftarrow \text{normalize}(\tilde{Q}_i(x_i)) \qquad\qquad\qquad\qquad\qquad\qquad \triangleright$ e.g. softmax

7: **end while**

---

## 4 Convolutional CRFs

The convolutional CRFs (ConvCRFs) supplement FullCRFs with a conditional independence assumption. We assume that the label distribution of two pixels $i, j$ are conditionally independent, if for the Manhattan distance $d$ holds $d(i, j) > k$. We call the hyperparameter $k$ filter-size.

This locality assumption is a very strong assumption. It implies that the pairwise potential is zero, for all pixels whose distance exceed $k$. This reduces the complexity of the pairwise potential greatly. The assumption can also be considered valid, given that CNNs are based on local feature processing and are highly successful. This makes the theoretical foundation of ConvCRFs very promising, strong and valid assumptions are the powerhouse of machine learning modelling.

### 4.1 Efficient Message Passing in ConvCRFs

One of the key contribution of this paper is to show that exact message passing is efficient in ConvCRFs. This eliminates the need to use the permutohedral lattice approximation, making highly efficient GPU computation and complete feature learning possible. Towards this goal we reformulate the message passing step to be a convolution with truncated Gaussian kernel and observe that this can be implemented very similar to regular convolutions in CNNs.

Consider an input $\mathbf{P}$ with shape $[bs, c, h, w]$ where $bs, c, h, w$ denote batch size, number of classes, input height and width respectively. For a Gaussian kernel $g$ defined by feature vectors $\mathbf{f}_1 \dots \mathbf{f}_d$, each of shape $[bs, h, w]$ we define its kernel matrix by

$$\mathbf{k}_g[b, dx, dy, x, y] := exp\left(-\sum_{i=1}^d \frac{|\mathbf{f}_i[b, x, y] - \mathbf{f}_i[b, x - dx, y - dy]|^2}{2\dot{\theta}_i^2}\right), \qquad (5)$$

where $\theta_i$ is a learnable parameter. For a set of Gaussian kernels $g_1 \dots g_s$ we define the merged kernel matrix $\mathbf{K}$ as $\mathbf{K} := \sum_{i=1}^s w_i \cdot g_i$. The result $Q$ of the combined message passing of all $s$ kernels is now given as:

$$Q[b, c, x, y] = \sum_{dx, dy \leq k} \mathbf{K}[b, dx, dy, x, y] \cdot \mathbf{P}[b, c, x + dx, y + dy]. \qquad (6)$$

This message passing operation is similar to standard 2d-convolutions of CNNs. In our case however, the filter values depend on the spatial dimensions $x$ and $y$. This is similar to locally connected layers (Chen et al., 2015b). Unlike locally connected layers (and unlike 2d-convolutions), our filters are however constant in the channel dimension $c$. One can view our operation as convolution over the dimension $c$ [1].

It is possible to implement our convolution operation by using standard CNN operations only. This however requires the data to be reorganized in GPU memory several times, which is a very slow process. Profiling shows that $90\,\%$ of GPU time is spend for the reorganization of data. We therefore opted to build a native low-level implementation, to gain an additional 10-fold speed up.

---

[1]Note that the operation refereed to as convolution in the context of NNs is actually known as *cross-correlation* in the signal processing community. The operation we want to implement however is a "proper" 1d-convolution. This operation is related to but not the same as 1d-convolution in NNs.

Efficient computation of our convolution can be implemented analogously to 2d-convolution (and locally connected layers). The first step is to tile the input $P$ in order to obtain data with shape $[bs, c, k, k, h, w]$. This process is usually referred to as *im2col* and the same as in 2d-convolutions (Chetlur et al., 2014). 2d-convolutions proceed by applying a batched matrix multiplication over the spatial dimension. We replace this step with a batched dot-product over the channel dimension. All other steps are the same.

## 4.2 ADDITIONAL IMPLEMENTATION DETAILS

For the sake of comparability we use the same design choices as FullCRFs in our baseline ConvCRF implementation. In particular, we use softmax normalization, the Potts model as well as the same hand-crafted gaussian features as proposed by Krähenbühl & Koltun (2011). Analogous to Krähenbühl & Koltun (2011) we also apply gaussian blur to the pairwise kernels. This leads to an increase of the effective filter size by a factor of $4$.

In additional experiments we investigate the capability of our CRFs to learn Gaussian features. Towards this goal we replace the input features $p_i$ of the smoothness kernel with learnable variables. Those variables are initialized to the same values as the hand-crafted version, but are adjusted as part of the training process. We also implement a learnable compatibility transformation using $1 \times 1$ convolution, following the ideas of Zheng et al. (2015).

## 5 EXPERIMENTAL EVALUATION

**Dataset:** We evaluate our method on the challenging PASCAL VOC 2012 (Everingham et al.) image dataset. Following the literature (Long et al., 2015; Wu et al., 2016; Chen et al., 2018; Zhao et al., 2017) we use the additional annotation provided by (Hariharan et al., 2011) resulting in $10\,582$ labelled images for training. Out of those images we hold back 200 images to fine-tune the internal CRF parameters and use the remaining $10\,382$ to train the unary CNN. We report our results on the $1464$ images of the official validation set.

**Unary:** We train a ResNet101 (He et al., 2016) to compute the unary potentials. We use the ResNet101 implementation provided by the PyTorch (Paszke et al., 2017) repository. A simple FCN (Long et al., 2015) is added on top of the ResNet to decode the CNN features and obtain valid segmentation predictions. The network is initialized using ImageNet Classification weights (Deng et al., 2009) and then trained on Pascal VOC data directly. Unlike many other projects, we do not train the network on large segmentation datasets such as MS COCO (Lin et al., 2014), but only use the images provided by the PASCAL VOC 2012 benchmark.

The CNN is trained for 200 epochs using a batch size of 16 and the adam optimizer (Kingma & Ba, 2014). The initial learning rate is set to $5 \times 10^{-5}$ and polynomially decreased (Liu et al., 2015; Chen et al., 2018) by multiplying the initial learning rate with $\left((1 - \frac{step}{max\_steps})^{0.9}\right)^2$. An $L_2$ weight decay with factor $5 \times 10^{-4}$ is applied to all kernel weights and 2d-Dropout (Tompson et al., 2015) with rate $0.5$ is used on top of the final convolutional layer. The same hyperparamters are also used for the end-to-end training.

The following data augmentation methods are applied: Random horizontal flip, random rotation ($\pm 10°$) and random resize with a factor in $(0.5, 2)$. In addition the image colours are jittered using random brightness, random contrast, random saturation and random hue. All random numbers are generated using a truncated normal distribution. The trained model achieves validation mIoU of $71.23\,\%$ and a train mIoU of $91.84\,\%$.

**CRF:** Following the literature (Krähenbühl & Koltun, 2011; Chen et al., 2018; Zheng et al., 2015; Lin et al., 2016), the mean-field inference of the CRF is computed for $5$ iterations in all experiments.

## 5.1 CONVCRFS ON SYNTHETIC DATA

To show the capabilities of Convolutional CRFs we first evaluate their performance on a synthetic task. We use the PASCAL VOC (Everingham et al.) dataset as a basis, but augment the ground-truth towards the goal to simulate prediction errors. The noised labels are used as unary potentials for the

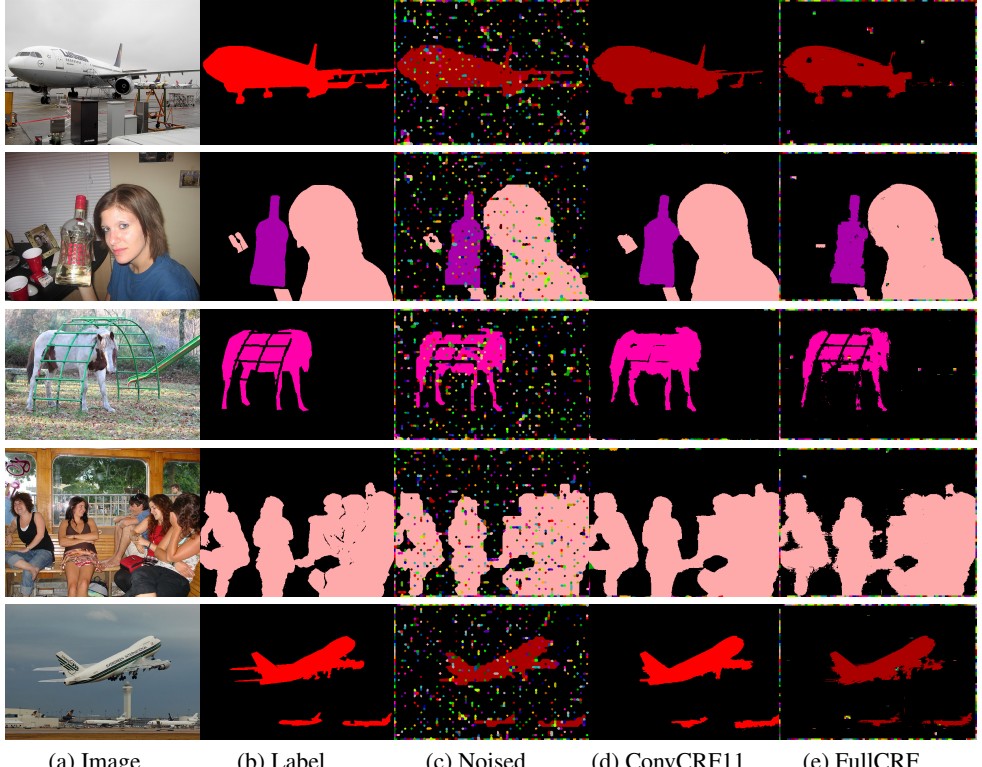

|     (a) Image     |     (b) Label     |     (c) Noised     |     (d) ConvCRF11     |     (e) FullCRF     |

Figure 1: Visualization of the synthetic task. Especially in the last example, the artefacts from the permutohedral lattice approximation can clearly be seen at object boundaries.

| Method | Unary | FullCRF | Conv5 | Conv7 | Conv11 | Conv13 | Conv15 |
|---|---|---|---|---|---|---|---|
| Speed [ms] | 68 | 647 | 7 | 13 | 26 | 34 | 45 |
| Accuracy [%] | 86.60 | 94.79 | 97.13 | 97.13 | 98.97 | 98.99 | 98.95 |
| mIoU [%] | 51.87 | 84.37 | 90.90 | 92.98 | 93.74 | 93.89 | 93.71 |

Table 1: Performance comparison of CRFs on the synthetic benchmark. The speed tests have been done on a Nvidia GeFore GTX 1080 Ti GPU and an Intel Xeon E5-2630 CPU. The images are processed in full resolution. ConvCRF utilize GPU computation while FullCRF inference is computed on CPU. Conv7 denotes a ConvCRF with filter size 7.

CRF, the CRF is then challenged to denoise the predictions. The output of the CRF is then compared to the original label of the Pascal VOC dataset.

Towards the goal of creating a relevant task, the following augmentation procedure is used: First the ground-truth is down-sampled by a factor of 8. Then, in low-resolution space random noise is added to the predictions and the result is up-sampled to the original resolution again. This process simulates inaccuracies as a result of the low-resolution feature processing of CNNs as well as prediction errors similar to the checkerboard issue found in deconvolution based segmentation networks (Gao et al., 2017; Odena et al., 2016). Some examples of the augmented ground-truth are shown in Figure 1.

In our first experiment we compare FullCRFs and ConvCRFs using the exact same parameters. To do this we utilize the hand-crafted Gaussian features. The remaining five parameters (namely $w^{(1)}$, $w^{(2)}$, as well as $\theta_\alpha$, $\theta_\beta$ and $\theta_\gamma$) are initialized to the default values proposed by Krähenbühl & Koltun (2011). Note that this gives FullCRFs a natural advantage. The performance of CRFs however is very robust with respect to these five parameters (Krähenbühl & Koltun, 2011).

The results of our first experiment are given in Table 1. It can be seen that ConvCRFs outperform FullCRFs significantly. This shows that ConvCRFs are structurally superior to FullCRFs. The better

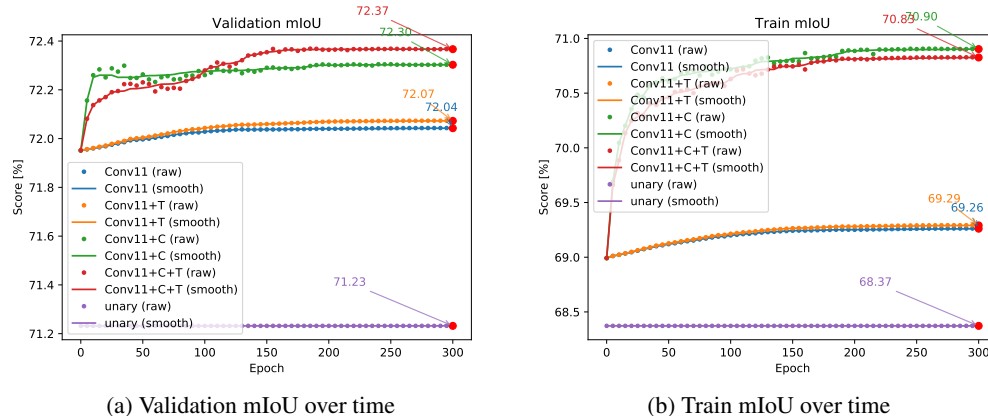

| | (a) Validation mIoU over time | (b) Train mIoU over time |

Figure 2: Training and validation mIoU over time for decoupled training. +C uses convolutions as compatibility transformation and +T learns the features for the smoothness kernel.

| Method | Unary | DeepLab | ConvCRF | Conv+T | Conv+C | **Conv+CT** |
|---|---|---|---|---|---|---|
| mIoU [%] | 71.23 | 72.02 | 72.04 | 72.07 | 72.30 | **72.37** |
| Accuracy [%] | 91.84 | 94.01 | 93.99 | 94.01 | 94.01 | **94.03** |
| train mIoU [%] | 68.37 | 68.61 | 69.26 | 69.29 | 70.90 | **70.83** |

Table 2: Performance comparison of CRFs on validation data using decoupled training. +C uses convolutions as compatibility transformation and +T learns the Gaussian features. The same unaries were used for all approaches, only the CRF code from DeepLab was utilized.

performance of ConvCRFs with the same parameters can be explained by our exact message passing, which avoids the approximation errors compared of the permutohedral lattice approximation. We provide a visual comparison in Figure 1 where ConvCRF clearly provide higher quality output. The FullCRF output shows approximation artefacts at the boundary of objects. In addition we note that ConvCRFs are faster by two orders of magnitude, making them favourable in almost every use case.

## 5.2 DECOUPLED TRAINING OF CONVCRFS

In this section we discuss our experiments on Pascal VOC data using a two stage training strategy. First the unary CNN model is trained to perform semantic segmentation on the Pascal VOC data. Those parameters are then fixed and in the second stage the internal CRF parameters are optimized with respect to the CNN predictions. The same unary predictions are used across all experiments, to reduce variants between runs.

Decoupled training has various merits compared to an end-to-end pipeline. Firstly it is very flexible. A standalone CRF training can be applied on top of any segmentation approach. The unary predictions are treated as a black-box input for the CRF training. In practice this means that the two training stages do not need to interface at all, making fast prototyping very easy. Additionally decoupled training keeps the system interpretable. Lastly, piecewise training effectively tackles the vanishing gradient problem (Bengio et al., 1994), which is still an issue in CNN based segmentation approaches (Wu et al., 2016). This leads to overall faster, more robust and reliable training.

For our experiments we train the CRF models on the 200 held-out images from the training set and evaluate the CRF performance on the 1464 images of the official Pascal VOC dataset. We compare the performance of the ConvCRF with filter size 11 to the unary baseline results as well as a FullCRF trained following the methodology of DeepLab (Chen et al., 2018).

We report our results in Table 2, the training curves are visualized in Figure 2. In all experiments, applying CRFs boost the performance considerably. The experiments also confirm the observation of Section 5.1, that ConvCRF perform slightly better than FullCRFs. We also observe that the ConvCRF

| Method | Unary | CRFasRNN | ConvCRF |
|---|---|---|---|
| mIoU [%] | 71.23 | 72.12 | 73.06 |
| Accuracy [%] | 93.80 | 94.08 | 94.27 |
| train mIoU [%] | 91.84 | 93.70 | 93.71 |

Table 3: Performance comparison of end-to-end trained CRFs.

implementation utilizing a learnable compatibility transformation as well as learnable Gaussian features performs best. Model output is visualized in Figure 3.

### 5.3 END-TO-END LEARNING WITH CONVCRFS

In this section we discuss our experiments using an end-to-end learning strategy for ConvCRFs. In end-to-end training the gradients are propagated through the entire pipeline. This allows the CNN and CRF model to co-adapt and therefore to produce the optimum output w.r.t the entire network. The down-side of end-to-end training is that the gradients need to be propagated through five iterations of the mean-field inference, resulting in vanishing gradients (Zheng et al., 2015).

We train our network for 250 epochs using a training protocol similar to CRFasRNN (Zheng et al., 2015). Zheng et al. propose to first train the unary potential until convergence and then optimizing the CRF and CNN jointly. Like Zheng et al. (2015) we greatly reduce the learning rate to $10^{-10}$ during the second training stage. We use a batch size of 16 for the first and 8 for the second training stage. In this regard we differ from Zheng et al. (2015), who proposes to reduce the batch size to 1 for the second training stage.

The entire training process takes about 30 hours using four 1080Ti GPUs in parallel. We believe that the fast training and inference speeds will greatly benefit and ease future research using CRFs. We compare our ConvCRF to the approach proposed in CRFasRNN (Zheng et al., 2015) and report the results in Table 3. Overall we see that ConvCRF slightly outperforms CRFasRNN at a much higher speed.

## 6 CONCLUSION

In this work we proposed Convolutional CRFs, a novel CRF design. Adding the strong and valid assumption of conditional independence enables us to remove the permutohedral lattice approximation. This allows us to implement the message passing highly efficiently on GPUs as convolution operations. This increases training and inference speed by two orders of magnitude. In addition we observe a modest accuracy improvement when computing the message passing exactly. Our method also enables us to easily train the Gaussian features of the CRF using backpropagation.

In future work we will investigate the potential of learning Gaussian features further. We are also going to examine more sophisticated CRF architectures, towards the goal of capturing context information even better. Lastly we are particularly interested in exploring the potential of ConvCRFs in other structured applications such as instance segmentation, landmark recognition and weakly supervised learning.

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

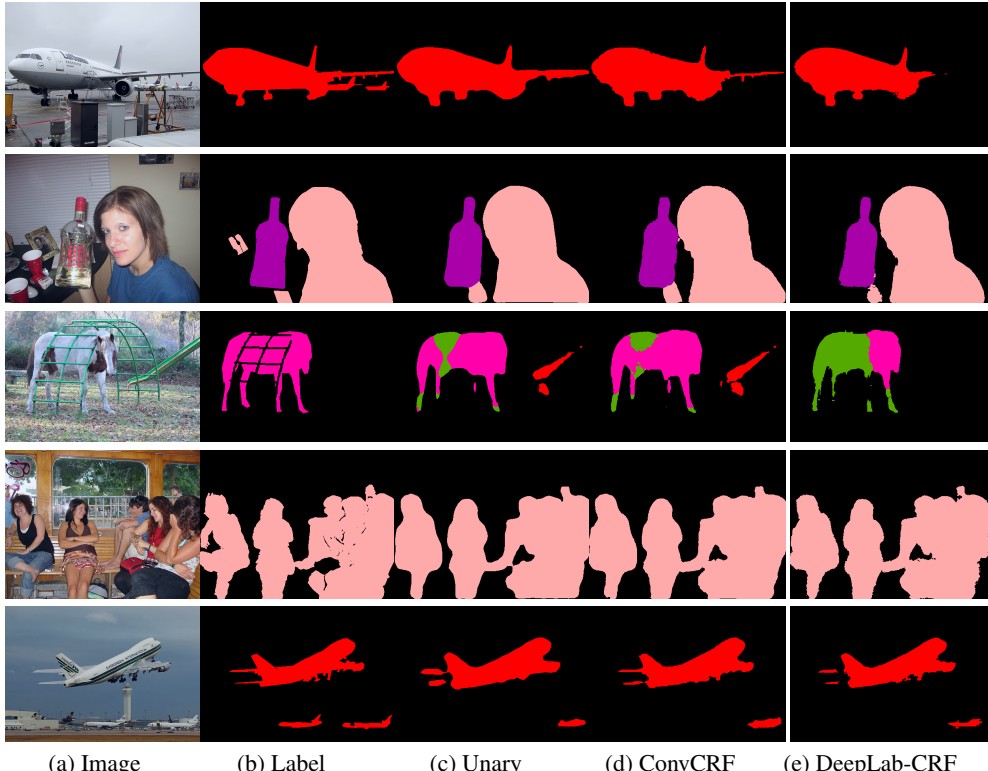

|         |           |           |             |                  |
| :-----: | :-------: | :-------: | :---------: | :--------------: |
| (a) Image | (b) Label | (c) Unary | (d) ConvCRF | (e) DeepLab-CRF |

Figure 3: Visualization of results on Pascal VOC data using a decoupled training strategy. Examples 2 and 4 depict failure cases, in which the CRFs are not able to improve the unary.

Siddhartha Chandra and Iasonas Kokkinos. Fast, exact and multi-scale inference for semantic image segmentation with deep gaussian crfs. In *European Conference on Computer Vision*, pp. 402–418. Springer, 2016.

Liang-Chieh Chen, George Papandreou, Iasonas Kokkinos, Kevin Murphy, and Alan L. Yuille. Semantic image segmentation with deep convolutional nets and fully connected crfs. *International Conference on Learning Representations*, 2015a. URL https://arxiv.org/pdf/1412.7062.pdf.

Liang-Chieh Chen, George Papandreou, Florian Schroff, and Hartwig Adam. Rethinking atrous convolution for semantic image segmentation. *arXiv preprint arXiv:1706.05587*, 2017.

Liang-Chieh Chen, George Papandreou, Iasonas Kokkinos, Kevin Murphy, and Alan L Yuille. Deeplab: Semantic image segmentation with deep convolutional nets, atrous convolution, and fully connected crfs. *IEEE transactions on pattern analysis and machine intelligence*, 40(4):834–848, 2018.

Yu-hsin Chen, Ignacio Lopez-Moreno, Tara N Sainath, Mirkó Visontai, Raziel Alvarez, and Carolina Parada. Locally-connected and convolutional neural networks for small footprint speaker recognition. In *Sixteenth Annual Conference of the International Speech Communication Association*, 2015b.

Sharan Chetlur, Cliff Woolley, Philippe Vandermersch, Jonathan Cohen, John Tran, Bryan Catanzaro, and Evan Shelhamer. cudnn: Efficient primitives for deep learning. *CoRR*, abs/1410.0759, 2014. URL http://arxiv.org/abs/1410.0759.

J. Deng, W. Dong, R. Socher, L.-J. Li, K. Li, and L. Fei-Fei. ImageNet: A Large-Scale Hierarchical Image Database. In *CVPR09*, 2009.

M. Everingham, L. Van Gool, C. K. I. Williams, J. Winn, and A. Zisserman. The PASCAL Visual Object Classes Challenge 2012 (VOC2012) Results. http://www.pascal-network.org/challenges/VOC/voc2012/workshop/index.html.

Hongyang Gao, Hao Yuan, Zhengyang Wang, and Shuiwang Ji. Pixel deconvolutional networks. *CoRR*, abs/1705.06820, 2017. URL http://arxiv.org/abs/1705.06820.

Bharath Hariharan, Pablo Arbeláez, Lubomir Bourdev, Subhransu Maji, and Jitendra Malik. Semantic contours from inverse detectors. In *Computer Vision (ICCV), 2011 IEEE International Conference on*, pp. 991–998. IEEE, 2011.

Kaiming He, Xiangyu Zhang, Shaoqing Ren, and Jian Sun. Deep residual learning for image recognition. In *Proceedings of the IEEE conference on computer vision and pattern recognition*, pp. 770–778, 2016.

Xuming He and Richard S Zemel. Learning hybrid models for image annotation with partially labeled data. In *Advances in Neural Information Processing Systems*, pp. 625–632, 2009.

Diederik P Kingma and Jimmy Ba. Adam: A method for stochastic optimization. *arXiv preprint arXiv:1412.6980*, 2014.

Philipp Krähenbühl and Vladlen Koltun. Efficient inference in fully connected crfs with gaussian edge potentials. In *Advances in neural information processing systems*, pp. 109–117, 2011.

Philipp Krähenbühl and Vladlen Koltun. Efficient inference in fully connected crfs with gaussian edge potentials (code). graphics.stanford.edu/projects/densecrf/densecrf.zip, 2011.

Philipp Krähenbühl and Vladlen Koltun. Parameter learning and convergent inference for dense random fields. In *International Conference on Machine Learning*, pp. 513–521, 2013.

Alex Krizhevsky, Ilya Sutskever, and Geoffrey E Hinton. Imagenet classification with deep convolutional neural networks. In *Advances in neural information processing systems*, pp. 1097–1105, 2012.

Qizhu Li, Anurag Arnab, and Philip HS Torr. Weakly-and semi-supervised panoptic segmentation. In *Proceedings of the European Conference on Computer Vision (ECCV)*, pp. 102–118, 2018.

Guosheng Lin, Chunhua Shen, Anton Van Den Hengel, and Ian Reid. Efficient piecewise training of deep structured models for semantic segmentation. In *Proceedings of the IEEE Conference on Computer Vision and Pattern Recognition*, pp. 3194–3203, 2016.

Tsung-Yi Lin, Michael Maire, Serge Belongie, James Hays, Pietro Perona, Deva Ramanan, Piotr Dollár, and C Lawrence Zitnick. Microsoft coco: Common objects in context. In *European conference on computer vision*, pp. 740–755. Springer, 2014.

Wei Liu, Andrew Rabinovich, and Alexander C Berg. Parsenet: Looking wider to see better. *arXiv preprint arXiv:1506.04579*, 2015.

Jonathan Long, Evan Shelhamer, and Trevor Darrell. Fully convolutional networks for semantic segmentation. In *Proceedings of the IEEE conference on computer vision and pattern recognition*, pp. 3431–3440, 2015.

John Nickolls, Ian Buck, Michael Garland, and Kevin Skadron. Scalable parallel programming with cuda. In *ACM SIGGRAPH 2008 classes*, pp. 16. ACM, 2008.

Hyeonwoo Noh, Seunghoon Hong, and Bohyung Han. Learning deconvolution network for semantic segmentation. In *Proceedings of the IEEE International Conference on Computer Vision*, pp. 1520–1528, 2015.

Augustus Odena, Vincent Dumoulin, and Chris Olah. Deconvolution and checkerboard artifacts. *Distill*, 1(10):e3, 2016.

Adam Paszke, Abhishek Chaurasia, Sangpil Kim, and Eugenio Culurciello. Enet: A deep neural network architecture for real-time semantic segmentation. *arXiv preprint arXiv:1606.02147*, 2016.

Adam Paszke, Sam Gross, Soumith Chintala, Gregory Chanan, Edward Yang, Zachary DeVito, Zeming Lin, Alban Desmaison, Luca Antiga, and Adam Lerer. Automatic differentiation in pytorch. In *NIPS-W*, 2017.

Olaf Ronneberger, Philipp Fischer, and Thomas Brox. U-net: Convolutional networks for biomedical image segmentation. In *International Conference on Medical image computing and computer-assisted intervention*, pp. 234–241. Springer, 2015.

David E Rumelhart, Geoffrey E Hinton, and Ronald J Williams. Learning representations by back-propagating errors. *nature*, 323(6088):533, 1986.

Alexander G. Schwing and Raquel Urtasun. Fully connected deep structured networks. *CoRR*, abs/1503.02351, 2015. URL http://arxiv.org/abs/1503.02351.

K. Simonyan and A. Zisserman. Very deep convolutional networks for large-scale image recognition. In *International Conference on Learning Representations*, 2015.

Meng Tang, Federico Perazzi, Abdelaziz Djelouah, Ismail Ben Ayed, Christopher Schroers, and Yuri Boykov. On regularized losses for weakly-supervised cnn segmentation. *arXiv preprint arXiv:1803.09569*, 2018.

Marvin Teichmann, Michael Weber, Marius Zoellner, Roberto Cipolla, and Raquel Urtasun. Multinet: Real-time joint semantic reasoning for autonomous driving. *arXiv preprint arXiv:1612.07695*, 2016.

Jonathan Tompson, Ross Goroshin, Arjun Jain, Yann LeCun, and Christoph Bregler. Efficient object localization using convolutional networks. In *Proceedings of the IEEE Conference on Computer Vision and Pattern Recognition*, pp. 648–656, 2015.

Bill Triggs and Jakob J Verbeek. Scene segmentation with crfs learned from partially labeled images. In *Advances in neural information processing systems*, pp. 1553–1560, 2008.

Raviteja Vemulapalli, Oncel Tuzel, Ming-Yu Liu, and Rama Chellapa. Gaussian conditional random field network for semantic segmentation. In *Proceedings of the IEEE Conference on Computer Vision and Pattern Recognition*, pp. 3224–3233, 2016.

Zifeng Wu, Chunhua Shen, and Anton van den Hengel. Wider or deeper: Revisiting the resnet model for visual recognition. *CoRR*, abs/1611.10080, 2016. URL http://arxiv.org/abs/1611.10080.

Fisher Yu and Vladlen Koltun. Multi-scale context aggregation by dilated convolutions. *arXiv preprint arXiv:1511.07122*, 2015.

Hengshuang Zhao, Jianping Shi, Xiaojuan Qi, Xiaogang Wang, and Jiaya Jia. Pyramid scene parsing network. In *IEEE Conf. on Computer Vision and Pattern Recognition (CVPR)*, pp. 2881–2890, 2017.

Shuai Zheng, Sadeep Jayasumana, Bernardino Romera-Paredes, Vibhav Vineet, Zhizhong Su, Dalong Du, Chang Huang, and Philip HS Torr. Conditional random fields as recurrent neural networks. In *Proceedings of the IEEE International Conference on Computer Vision*, pp. 1529–1537, 2015.

