# OpenReview forum: "Convolutional CRFs for Semantic Segmentation"
_ICLR.cc/2019/Conference_

### Official Review · AnonReviewer1 · 2018-10-30
**Using convolution in CRFs to increase inference and training time speed**

**Rating:** 6
**Confidence:** 4

**Review:**

The paper is well written with many relevant references and easy to read. Some points that need clarification and mentioned below.
The main points of this paper are the use of the convolution operator to perform the message passing mean field inference. Using this operation allows us to get away from the permutohedral lattice and yet allows speed up of 100x. This also means, that training will be able to done faster. Besides this the training parameters can also be learnt. These are the main contributions. The denoising task experiment shows positive results. The idea could be used in the future by others looking for faster model inference and training.

If a Manhattan distance d is used i.e. dx,dy<k in equation (6), why is this a FullCRF? It seems like the new CRF is no longer a fully connected one.

Page 5, first paragraph describing how the reorganization in the GPU is avoided is not very clear. It would be helpful to a reader to have more details and explanations about this.

It is not clear from the experimental results how much improvement allowing to train the CRF parameters gets or might get. Comparing to the Deeplab results etc for the non-trained case, the non-trained model still seems to be performing competitively. Table 2 of Table 3 does not really bring out the advantage of training. The +C, +T, +CT don't seem to be hugely different in terms of validation metrics. Note that Table 3 does not mention other models that might not be trained (assuming that those results are in Table 2) but the text also mentions that the training is not completely fair.

In section 5, Unary, it is mentioned that the network is not trained on larger datasets like other work, why?
And under CRF, what does iterations are unrolled mean?

In section 5.1, why does the random flipping help in simulating inaccuracies?


Minor points:
Abstract: Add space after "GPUs.".
Would be good to define what Q, *, ' indicate in paragraph 4, page 2.
"hight" -> "height" in section 4.1

---

> ### Author Response · Authors · 2018-11-26
> **Re: Using convolution in CRFs to increase inference and training time speed**
>
> >> If a Manhattan distance d is used i.e. dx,dy<k in equation (6), why is this a FullCRF? It seems like the new CRF is no longer a fully connected one.
>
> Yes, this is why we call out method ConvolutionalCRF. FullCRF refers to "Efficient Inference in Fully Connected CRFs with Gaussian Edge Potentials", Krähenbühl & Vladlen Koltun (2012).
>
> >> It is not clear from the experimental results how much improvement allowing to train the CRF parameters gets or might get. Comparing to the Deeplab results etc for the non-trained case, the non-trained model still seems to be performing competitively.
>
> Yes, the performance (IoU) of DeepLab CRF is very comparable to the performance of ConvCRF. This is not surprising as ConvCRF was designed to be as close as possible to FullCRF. The goal is to show that we achieve similar performance at a much larger speed. The same holds for the trainable potentials. The main goal is to show that our approach is able to train potentials. Trainable potentials open up a large design space for future research where our FullCRF based model can serve as an initial baseline.
>
> >> In section 5, unary, it is mentioned that the network is not trained on larger datasets like other work, why?
>
> The single digit performance gain by training on external datasets does not justify the large amount of additional GPU time required for those experiments. This also makes our experiments much easier to reproduce for individuals and people in academia.
>
> >> And under CRF, what does iterations are unrolled mean?
>
> In our initial implementation we have implemented RNN like a five layer CNN which shares weights. This is not true in our newest (published) implementation, we have removed the statement. (Both implementation behave the same, given same weights).
>
> >> In section 5.1, why does the random flipping help in simulating inaccuracies?
>
> With random flipping we meant that some predictions are randomly changed. We have changed to wording to "random noise is added to the predictions" as the word flipping might be misleading.

---

### Official Review · AnonReviewer3 · 2018-11-02
**Confusing notation, insufficient analysis, main contribution unclear**

**Rating:** 4
**Confidence:** 4

**Review:**

The authors propose an efficient method to perform message passing on a truncated Gaussian kernel CRF. The main contributions are the definition of a specific form of truncated Gaussian kernel that allows for fast message passing via convolutions, and the implementation of such parallelized message passing on GPU.

In my opinion, the paper fails to convey the main idea in a clear and precise manner, the notation is mixed and often confusing, furthermore there are a number of sentences that should be rephrased to be less sensationalist, or removed. The experiments seem to show performance in par with the FullCRF on decoupled training, which seem in contrast with the much bigger performance gain of the first experiment on syntetic data. No discussion has been provided as to the possible reasons of this performance gap, although the experimental settings appear to be similar. Finally, in the last experiments with end-to-end training the authors report a performance improvement over CRFasRNN, a 3 years old paper that is far away in terms of performance with the current SOTA on Pascal VOC. The authors base on a different network than that of the CRFasRNN baseline (i.e., the difference is not only in the CRF implementation, but rather the whole network before the CRF in the proposed method), it is therefore difficult to say whether the performance improvement is due to the ResNet101 + FCN unary potentials, which is not a contribution of this manuscript, or to the proposed CRF. In general, I believe that the considerable speed gain of the proposed method might be enough to justify a publication, but the paper should be phrased in that sense if that was the intention of the authors. It is unclear to me whether the main contribution they claim is segmentation performance (IoU) or speed or both. The main contributions of this work should be stated clearly, and the modelling differences w.r.t. the FullCRF model that they aim to improve should be more explicit in the text rather than let to the reader to infer comparing the formulas.

On these grounds, I suggest a major revision of the paper and I don't recommend publication at this stage.



MAJOR

1) I firmly advocate against making strong claims, unless supported by solid proofs. I strongly recommend to rephrase, if not remove, exaggerate claims such as:

a) "[deep networks] lack the capability to utilize context information and cannot model interactions between predictions directly". This is simply not true. Any CNN with enough layers will exploit contextual information. Furthermore, any autoregressive model will model the interaction between predictions directly. See e.g., "RiFCN: Recurrent Network in Fully Convolutional Network for Semantic Segmentation of High Resolution Remote Sensing Images" by Mou et Al., "ReSeg: A Recurrent Neural Network-based Model for Semantic Segmentation" by Visin et Al., or "Predicting Deeper into the Future of Semantic Segmentation" by Luc et Al. for video semantic segmentation.

b) "CRF inference is two orders of magnitude slower than CNN inference": this, of course, depends on the kind of CRF.

c) "The long training times of the current generation of CRFs also make more in-depth research and experiments with such structured models impractical": again, not true. While it's true that CRFs tend to be slow, research with such models is not impractical and indeed there are papers that focus exactly on that (among the others, some of the ones cited in this manuscript).

d) "we propose to add the strong and valid assumption of conditional independence": as with every assumption, this is an approximation. I wouldn't claim it to be valid nor invalid, as it is simply a modeling choice.

e) "Predictions are pixel-wise and conditionally independendent (given the common feature base of nearby pixels). Structured knowledge and background context is ignored in these models.". It's unclear what is meant by "structured knowledge", but to my best understanding this sentence is misleading or wrong. Background context is considered by CNN-based models, as well as the general structure in the image.

f) "In the context of semantic segmentation most CRF based approaches are based on the Fully Connected CRF". CRFs have been used much earlier than 2011, before Fully Connected CRF was published.

g) "This makes the theoretical foundation of ConvCRF very promising, strong and valid assumptions are the powerhouse of machine learning modelling." The authors here propose a logical, but quite obvious, approximation, i.e., to constrain the CRF to model dependencies in a local neighborhood. This is the same implicit assumption of many other Gaussian kernels and algorithms for approximated inference. For how it's surely valid, and possibly strong, I don't see how it can make a "theoretical foundation". The self-congratulatory closure is unnecessary, and inappropriate.



2) While CRFs have been used for a long time on visual data, the citations in this work focus mostly on the last few years. I suggest to add at least one of the following:
* “Discriminative fields for modeling spatial dependencies in natural images" 2003
* "Multiscale conditional random fields for image labelling" 2004
* “Textonboost: Joint appearance, shape and context modeling for mulit-class object recognition and segmentation,” 2006

It could also be beneficial to the reader to add to the references broad overview works, such as "An Introduction to Conditional Random Fields" by Charles Sutton and Andrew McCallum, 2011, and/or "Structured prediction and learning in computer vision" by Nowozin and Lampert, 2011.



3) Line 5 of the algorithm adds the unary potentials at each iteration of message passing. Can you elaborate on the motivation behind this choice? The algorithm is already initialized with such potentials, and to my best knowledge unary potentials are not usually added in the mean field message passing loop. It would be interesting to compare the performance of the algorithm with and without this addition.



4) Sec4.1 reports that the filters are constant over the channel dimension c and that, in other words, this can be seen as applying the convolution over the dimension c. I fail to understand this sentence. My understanding from the formula is that the same kernel is applied to all the channels of the input, i.e., the channels of the input fed to the CRF are all processed in the same way. This should be explained clearly and the reasoning behind this choice should be explained as well. Furthermore, if I am not mistaken the authors learn a different filter in each position *of the input* rather than reusing the same filters at every position. This choice should be clarified and discussed.



5) Regarding the implementation, is there a reason not to apply a convolution with a flipped kernel to compute the cross-correlation? Also, IIRC if the kernel is symmetric (as should be for a Gaussian kernel) convolution and cross-correlation are the same.



6) The notation is often ambiguous and at times unnecessarely heavy. I strongly recommend to go over the manuscript and use a consistent notation, making sure that every element of the notation is introduced before or right after it's used. In particular,

a) Sec3: in the text, a segmentation instance is referred to as X, while in the formula as \hat x. \tilde I is never introduced. k_{\alpha} is defined but I believe never used (I suggest to drop the name if there is no reference to it). Is there a reason to drop the subscript G in the FullCRF pairwise potential? Or conversely, is there a reason to have it everywhere else? Furthermore, there doesn't seem to be a difference between k_G, k_g and g, I suggest to use only one consistent notation to refer to the Gaussian kernels throughout the paper. It's also unclear if the I superscript is needed for the feature vectors.

b) Sec4.1, The shape of the Gaussian kernels is the same as that of the input. I believe that the input in this context refers to a patch and not to the whole image. If so, this should be specified, otherwise the dimensions of the kernel should be referred to with a different letter than those of the input.

c) Sec4.1, I believe dx and dy refer to the in-kernel displacement. Their semantic is not clear from the text and should be defined properly.

d) Sec4.1, the feature vectors are defined in the text as f_1..f_d. The formula of the kernel uses f_i^{(d)} instead. It's unclear what the superscripts stands for and whether it is actually useful or redundant.

e) Sec4.1, x and y are not defined, I suspect they refer to the position of the pixel, which was previously encoded as p_i and p_j. Once again, the notation should be consistent across the manuscript.

f) In the definition of the convCRFs, w is used for the width of the input, w_i for the weights of the kernels. In the FullCRF, w^{(1)} and w^{(2)} for the potentials, w^{(m)} for the sum over the kernels. k is used for the kernel dimension, k_G to refer to the kernel itself, as well as g. The notation could be made less ambiguous and consistent (superscript vs subscript semantics).

g) Sec3, the number of pixel is defined as n but in Formula 2 N is used instead.

g) Vectors and matrices should be bold-face. The use of capital letters for constants might also improve the readability of the manuscript.



7) The experiments with the Conv (ConvCRF?) variants of Table 2 are not discussed in the text.



8) Although Sec5.2 concludes with "The experiments also confirm the observation of Sec5.1, that ConvCRF performs slightly better than FullCRF", Sec5.1 reported that "it can be seen that ConvCRFs outperform FullCRFs significantly". The authors should decide whether the results are slightly better or outperform the baseline. In general a in-depth discussion on the performance of the algorithm is missing.



9) Sec5.3, it's unclear what this sentence means "introduce an auxiliary unary loss to counterbalance the vanishing gradient problem". If such a term has been added, it should be reported in a formula and it's effectiveness should be supported by experimental data.



MINOR

m1) In the related work, the sentence "transposed convolution layers are applied at the end of the prediction pipeline ot produce high-resolution output" seems to suggest these are always applied, while many recent methods rely on bilinear upsampling to recover the original resolution. Please rephrase it accordingly.

m2) In Parameter learning in CRF: "the idea utilizes, that for the message passing the identity .. is valid." This sentence doesn't make any sense to me. Is it possible it is a leftover?

m4) In Sec3, the features vectors [...] may depend on the input image I. I am confused as to when they might be independent of the image. Can you elaborate on that?

m5) In Sec3, it's unclear to me what the vertical bar in the Pot model stands for. I believe the correct formula should be 1_{[xi != xj]}.

m6) In mean field inference, Algorithm 1 does not refer to FullCRFs.

m7) End of page 6, "Note that this gives FullCRFs a natural advantage. The performance of CRFs however is very robust [...]". Why is this an advantage for FullCRFs? How does that relate to the following sentence?

m8) Sec4.1, the authors claim that one of the key contribution of the paper is that exact message passing is efficient. Given the locality assumption, message passing is approximate - which is also why it's efficient. The authors could instead argue that using convolutions is faster and possibly leads to better final performance than using the permutohedral lattice approximation (although it's unclear whether this is the case from the experiments), with proper reference to compelling results in this direction.

Finally, a few typos:
* Abstract, space missing after GPUs
* Introduction, Convolutional Neuronal -> Convolutional Neural
* Introduction, order of magnitude slower then -> than
* Introduction, to slow -> too slow
* Parameter learning in CRF: missing space before proposed to use gradient descent
* Parameter learning in CRF: gradient decent -> descent
* Parameter learning in CRF: extra comma after "another advantage of this method is"
* Sec 3: "weighted sum of Gaussian kernels", the apex of the second should be "M" I believe.
* Sec 3: "can be chosen arbitrary" -> arbitrarily
* Sec 5.2, beginning of page 8: then -> than

---

> ### Author Response · Authors · 2018-11-26
> **Adressing Notation**
>
> I heavily disagree that my notation is confusion or incorrect. I am main using well established conventions and notation and I am quite thorough in defining open parameters and objects. Running indices in sums and matrices (like $i$ in $sum_{i} = i*i$) as well as explicit function arguments (like $x$ in $f(x) = x**2$) do not need to be further defined, since they are self contained in the equation.
>
> >> a) Sec3: in the text, a segmentation instance is referred to as X, while in the formula as \hat x. \tilde I is never introduced.
>
> No, $X$ is a random variable (as defined in the text) and $\hat x$ its realization. This follows well established  conventions from statistic and probability theory. The notation $P(X=\hat x | \tilde I = I) = ... $, is well understood across disciplines, $\hat x$ does not need to be further defined in this context. Analogously, since I define $I$ in the text it is clear that $\tilde I$ is a random variable which produces images. There is no disambiguity here. Also $\tilde I$ is only used in this equation for the sake of completeness.
>
> >> b) Sec4.1, The shape of the Gaussian kernels is the same as that of the input. I believe that the input in this context refers to a patch and not to the whole image. If so, this should be specified, otherwise the dimensions of the kernel should be referred to with a different letter than those of the input.
>
> The input refers to the whole image. The shape of the Gaussian kernel is not discussed in the section. I think you mean the shape feature vectors $f_1 ... f_d$ which define the kernel. Those feature vectors have the same shape as the input image (apart from the channel dimension). The notation deliberately emphasizes this and the relation is correct.
>
> >> c) Sec4.1, I believe dx and dy refer to the in-kernel displacement. Their semantic is not clear from the text and should be defined properly.
>
> I heavily disagree with this statement. The variables $dx$, $dy$ are running indices in a sum (eq. 6) or matrix definition (eq. 7) respectively. I have never seen a work where running indices are further defined in text since they are self-contained in the formula. It is true that traditionally most people would use $i$ and $j$ in this place. I choose $dx$, $dy$ as names for the variables to give the equations more semantic and help the reader understand why the equation follows this particular formula.
>
> >> d) Sec4.1, the feature vectors are defined in the text as f_1..f_d. The formula of the kernel uses f_i^{(d)} instead. It's unclear what the superscripts stands for and whether it is actually useful or redundant.
>
> The superscript was supposed to emphasize that $f_i$ is a vector. I do agree that this superscript might cause more harm then good, so it is removed. Vectors are now printed in bold instead.
>
> >> e) Sec4.1, x and y are not defined, I suspect they refer to the position of the pixel, which was previously encoded as p_i and p_j. Once again, the notation should be consistent across the manuscript.
>
> Again, $x$ and $y$ are running indices in a matrix definition. As such they are self contained. Also $p_i$ and $p_j$ are values of the smoothness kernel while $x$ and $y$ refer to coordinates in feature space. I think that it is useful to differentiate between these.

---

> > ### Author Response · Authors · 2018-11-26
> > **More on Notation**
> >
> > >> a') k_{\alpha} is defined but I believe never used (I suggest to drop the name if there is no reference to it). Is there a reason to drop the subscript G in the FullCRF pairwise potential? Or conversely, is there a reason to have it everywhere else? Furthermore, there doesn't seem to be a difference between k_G, k_g and g, I suggest to use only one consistent notation to refer to the Gaussian kernels throughout the paper.
> >
> > For a gaussian kernel $g$, $k_g$ denotes its kernel matrix (this relation is explicitly stated in the text). Analogously $k_{\alpha}$ is the kernel matrix for the kernel $\alpha$ and $k_G$ for the kernel $G$. We don't think that it is useful to explicitly state this, since this is how subscripting works. The objects $k_g$ and $g$ are isomorphic but they are not quite the same, that is why we like to distinguish between them.
> >
> > The kernel matrix $k_{\alpha}$ is widely used in FullCRF and related work, that is why we have explicitly named it. With $G$ we denote a gaussian kernel as used in FullCRFs, with $g$ a (truncated) gaussian kernel for ConvCRF (which follows our locality assumption). Mathematically this is correct, since we have defined $g$ and $G$ in the corresponding sections. However I can see that all of this might be to much notational sugar. We have therefore opted to remove $k_{\alpha}$ and denote both kernels as $g$.
> >
> > >> f) In the definition of the ConvCRFs, w is used for the width of the input, w_i for the weights of the kernels. In the FullCRF, w^{(1)} and w^{(2)} for the potentials, w^{(m)} for the sum over the kernels. k is used for the kernel dimension, k_G to refer to the kernel itself, as well as g. The notation could be made less ambiguous and consistent (superscript vs subscript semantics).
> >
> > I have already discussed the difference between the kernel matrix $k_G$, the kernel $G$ and the (other) kernel $g$. The weights $w^{(1)}$ is $w^{(m)}$ with $m=1$, same holds for $w^{(2)}$. In the formula we use $m$, since the general concept holds for any $m$. In addition  $w^{(1)}$ and $w^{(2)}$ are explicitly given since FullCRF only uses two kernels and we find it useful to write the unrolled formulation down. I don't think there is any confusion between the width $w$ in section 4.1 and the weights $w^{(m)}$ in section 3.
> >
> > >> g) Sec3, the number of pixel is defined as n but in Formula 2 N is used instead.
> >
> > Good spot, fixed.
> >
> > >> h) Vectors and matrices should be bold-face. The use of capital letters for constants might also improve the readability of the manuscript.
> >
> > Done.

---

> ### Author Response · Authors · 2018-11-26
> **Adressing Mayor Points**
>
> >> a) "[deep networks] lack the capability to utilize context information and cannot model interactions between predictions directly". This is simply not true. Any CNN with enough layers will exploit contextual information. Furthermore, any autoregressive model will model the interaction between predictions directly. See e.g., "RiFCN: Recurrent Network in Fully Convolutional Network for Semantic Segmentation of High Resolution Remote Sensing Images" by Mou et Al., "ReSeg: A Recurrent Neural Network-based Model for Semantic Segmentation" by Visin et Al., or "Predicting Deeper into the Future of Semantic Segmentation" by Luc et Al. for video semantic segmentation.
>
> The statement refers to "simple feed-forward CNN". I have reorganized the paragraph to make this clearer. That RNNs are capable of all those thinks is not surprising, ConvCRFs are an RNN model as well, after all.
>
> >> b) "CRF inference is two orders of magnitude slower than CNN inference": this, of course, depends on the kind of CRF.
>
> I have added a "typically" to the sentence. Of course it is possible to implement a very slow CNN and a degenerated CRF (i.e. one which models the identity). But discussing edge cases is not quite the point of this paragraph.
>
> >> c) "The long training times of the current generation of CRFs also make more in-depth research and experiments with such structured models impractical": again, not true. While it's true that CRFs tend to be slow, research with such models is not impractical and indeed there are papers that focus exactly on that (among the others, some of the ones cited in this manuscript).
>
> Impractical does not mean impossible. It is certainly possible to train a FullCRF based model, given enough time. Much faster training times will make it easier to iterate over ideas and perform in-depth analysis. We believe that our contribution will facilitate further research on CRF architectures.
>
> >> d) "we propose to add the strong and valid assumption of conditional independence": as with every assumption, this is an approximation. I wouldn't claim it to be valid nor invalid, as it is simply a modeling choice.
>
> I strongly disagree with the notion that assumptions are merely approximation. Assumptions are the structure we impose upon the model. Take CNNs for example, I strongly believe that they are so successfull because they do incooperate the assumptions of feature locality and transposition invariance into a dnn model. Would you argue that CNNs are an approximation for fully-connected NNs? I would also claim that both assumptions (feature locality and transposition invariance) are valid for most natural image processing tasks since I do not know any model which works well and does not incooperate those in some way or another. Since the locality implies the conditional independence we formulate, we can conclude that our conditional independence is also valid.
>
> >> e) "Predictions are pixel-wise and conditionally independendent (given the common feature base of nearby pixels). Structured knowledge and background context is ignored in these models.". It's unclear what is meant by "structured knowledge", but to my best understanding this sentence is misleading or wrong. Background context is considered by CNN-based models, as well as the general structure in the image.
>
> CNNs do not model a conditional relation between the predictions. The only way this structural knowledge is utilized is through a common feature base of nearby predictions. Beeing able to formulate those relation explicitly (i.e. as E ~ P(p_i=x|p_j=y)) opens further design space.
>
> >> f) "In the context of semantic segmentation most CRF based approaches are based on the Fully Connected CRF". CRFs have been used much earlier than 2011, before Fully Connected CRF was published.
>
> I have added the word *recent* to the sentence.
>
> >> g) "This makes the theoretical foundation of ConvCRF very promising, strong and valid assumptions are the powerhouse of machine learning modelling." The authors here propose a logical, but quite obvious, approximation, i.e., to constrain the CRF to model dependencies in a local neighborhood...
>
> If our contribution is so obvious, why have people kept using FullCRF model instead? Also, sometimes it just needs somebody to do the obvious think.

---

> > ### Author Response · Authors · 2018-11-26
> > **Adressing further points**
> >
> > >> 3) Line 5 of the algorithm adds the unary potentials at each iteration of message passing. Can you elaborate on the motivation behind this choice? The algorithm is already initialized with such potentials, and to my best knowledge unary potentials are not usually added in the mean field message passing loop. It would be interesting to compare the performance of the algorithm with and without this addition.
> >
> > In short, we do this because FullCRF "Efficient Inference in Fully Connected CRFs with Gaussian Edge Potentials" does the same. In general our approach is to follow this work as close as possible so that the experimental results highlight the advantages of our convolutional message passing. While I do agree that this would be an interesting study, we don't think that this paper is the right place to conduct it.
> >
> > >> My understanding from the formula is that the same kernel is applied to all the channels of the input, i.e., the channels of the input fed to the CRF are all processed in the same way
> >
> > Yes, it is.
> >
> > >> This should be explained clearly and the reasoning behind this choice should be explained as well. Furthermore, if I am not mistaken the authors learn a different filter in each position *of the input* rather than reusing the same filters at every position. This choice should be clarified and discussed.
> >
> > The reasoning is the same as 3: we do it like this, because FullCRF does this as well. While our implementation differs quite a bit the formulation was derived by using the formula given in FullCRF and formulate it in terms of convolutions. After accounting for approximation errors, our formula produces the same output as the lattice filtering. We explicitly verified this with test cases.
> >
> > >> 5) Regarding the implementation, is there a reason not to apply a convolution with a flipped kernel to compute the cross-correlation? Also, IIRC if the kernel is symmetric (as should be for a Gaussian kernel) convolution and cross-correlation are the same.
> >
> > Great spot, the relation should indeed hold. One additional issue to consider is that we need a fairly unusual padding, since we are not convolution over the dimension which needs padding. Efficient padding in convolutions is implemented during the lower-level im2col step. Also, im2col has been exposed to the python front-end of both PyTorch and Tensorflow in recent versions. (In the case of PyTorch in part thanks to our feedback). So it is now possible to implement eq 6, completely in python without the need to directly touch cuda.
> >
> > >> 7) The experiments with the Conv (ConvCRF?) variants of Table 2 are not discussed in the text.
> >
> > They are: Page 7 last paragraph and Page 8 first paragraph.
> >
> > >> 8) Although Sec5.2 concludes with "The experiments also confirm the observation of Sec5.1, that ConvCRF performs slightly better than FullCRF", Sec5.1 reported that "it can be seen that ConvCRFs outperform FullCRFs significantly". The authors should decide whether the results are slightly better or outperform the baseline. In general a in-depth discussion on the performance of the algorithm is missing.
> >
> > Sec5.2 discusses our experiments using a decoupled training for a semantic segmentation task. For this setting ConvCRFs have a slightly better score then FullCRFs. However the different is very modest. Thus we think that the experiment shows that ConvCRFs and FullCRFs are on par with respect to segmentation performance (IoU). In section 5.1 we discuss our experimental results for our synthetic denoising task. This is a different experiment and for this experiment we conclude that ConvCRFs do outperform the baseline. We believe that it is fine that different experiments on different tasks can yield different results. As an overall conclusion we descripe the performance improvement as "modest" in section 6.
> >
> >
> > >> 9) Sec5.3, it's unclear what this sentence means "introduce an auxiliary unary loss to counterbalance the vanishing gradient problem". If such a term has been added, it should be reported in a formula and it's effectiveness should be supported by experimental data.
> >
> > In the newest version of experiments we don't use an auxiliary loss anymore, since we are following the training protocol of CRFasRNN more closely. For the sake of completness: The total loss was given as $0.5 * loss_crf + 0.5 * loss_unary$, where $loss_unary$ is the loss computed on the unary (i.e. the same loss as used in sec5.1 and sec5.2) and $loss_crf$ is the loss computed on the crf output. We don't think that this was important enough to justify an ablation study. Using auxiliary loss is a common enough context which is often mentioned without explicitly discussion the formulation.

---

> ### Author Response · Authors · 2018-11-26
> **Adressing minor points**
>
> >> m1) In the related work, the sentence "transposed convolution layers are applied at the end of the prediction pipeline ot produce high-resolution output" seems to suggest these are always applied, while many recent methods rely on bilinear upsampling to recover the original resolution. Please rephrase it accordingly.
>
> Bilinear upsampling is a transposed convolution operation. Bilineary upsampling is obtained by choosing a specific parameter configuration. To the best of my knowledge all those methods backpropagate through the bilinear upsampling and they are using cudnn convolutions for it. Therefore I would argue it is correct to name a bilinear upsampling layer a transposed convolution layers.
>
> >> m2) In Parameter learning in CRF: "the idea utilizes, that for the message passing the identity .. is valid." This sentence doesn't make any sense to me. Is it possible it is a leftover?
>
> No, the sentence is quite crucial. I have rephrased it to: "The idea utilizes, that for the message passing the equation $(k_G * Q)' = k_G * Q'$ holds."
>
> >> m4) In Sec3, the features vectors [...] may depend on the input image I. I am confused as to when they might be independent of the image. Can you elaborate on that?
>
> The bilinear potential in "Efficient Inference in Fully Connected CRFs ..." does not depend on the image. So while it does not have to depend on the image the features of any meaningful CRF would depend on the Image. Also, the goal of this sentence is the opposite. I wanted to emphasize that $f$ is usually a function of $I$. In the literature this dependency is quite often omitted, which I find confusing at times.
>
>
> >> m5) In Sec3, it's unclear to me what the vertical bar in the Pot model stands for. I believe the correct formula should be 1_{[xi != xj]}.
>
> The vertical bars stand for absolute value, it indicates that the truth value should be interpreted as number. I have adapted your notation, but in my eyes both formulas ($1_{[xi != xj]}$ and |x_i != x_j| mean the same.
>
> >> m6) In mean field inference, Algorithm 1 does not refer to FullCRFs.
>
> Fixed.
>
> >> m7) End of page 6, "Note that this gives FullCRFs a natural advantage. The performance of CRFs however is very robust [...]". Why is this an advantage for FullCRFs? How does that relate to the following sentence?
>
> The advantage refers to the previous sentence: The remaining five parameters [...] are initialized to the default values proposed by Krähenbühl & Koltun. This is an advantage for FullCRFs since the defaults were choosen for the FullCRF architecture. The relation to the following sentence is, that this is "however" not a big deal overall, since any reasonable choice for those parameters yields a similar performance.
>
> >> m8) Sec4.1, the authors claim that one of the key contribution of the paper is that exact message passing is efficient. Given the locality assumption, message passing is approximate - which is also why it's efficient. The authors could instead argue that using convolutions is faster and possibly leads to better final performance than using the permutohedral lattice approximation (although it's unclear whether this is the case from the experiments), with proper reference to compelling results in this direction.
>
> The sentence reads: "exact message passing is efficient *in ConvCRFs*". This is correct, given a ConvCRF model as defined in section 4, using a filter with our locallity constrain, message passing is efficient. Also, the goal of the paragraph is not to spark a discussion about assumptions. It rather serves to communicate that our message passing can be performed without the high-dimensional filtering approximation, which is responsible for slowness and gradient issues in FullCRFs. I believe that the paragraph does this very well.
>
>
> >> Finally, a few typos:
>
> Thanks, all have been addressed.

---

> ### Author Response · Authors · 2018-11-26
> **Adressing general points.**
>
> >> In general, I believe that the considerable speed gain of the proposed method might be enough to justify a publication, but the paper should be phrased in that sense if that was the intention of the authors. It is unclear to me whether the main contribution they claim is segmentation performance (IoU) or speed or both.
>
> The main contribution of the paper is the considerable speed gain as well as the ability to learn the Gaussian parameters using backprop. We have stated this several times throughout the paper, e.g. in the abstract, introduction as well as conclusion.  We do not claimed that our method improves segmentation performance (IoU) significantly across experiments. We report the IoU scores to show that the performance of ConvCRFs is on par with FullCRFs while beeing two orders of magnitude faster.
>
> >> The main contributions of this work should be stated clearly, and the modelling differences w.r.t. the FullCRF model that they aim to improve should be more explicit in the text rather than let to the reader to infer comparing the formulas.
>
> I fully agree that the contributions and aim of the paper should be stated clearly. This is why I have formulated the goals and contributions of the paper several times in the abstract, introduction and conclusion. This is also the first time I got the feedback that our goals are not stated clearly, my overall impression is that most readers are able to easily grasp the key contributions our paper.
>
> >> In my opinion, the paper fails to convey the main idea in a clear and precise manner, the notation is mixed and often confusing, furthermore there are a number of sentences that should be rephrased to be less sensationalist, or removed.
>
> I respectfully disagree with both statements. My impression is that we might come from different schools of thought when it comes to both notation but also interpretation behind machine learning concepts. This might lead to misunderstanding of notation but also the exact interpretation and gravity of certain claims. See the discussion on mayor points and notation for more details.
>
> >> The authors base on a different network than that of the CRFasRNN baseline (i.e., the difference is not only in the CRF implementation, but rather the whole network before the CRF in the proposed method), it is therefore difficult to say whether the performance improvement is due to the ResNet101 + FCN unary potentials, which is not a contribution of this manuscript, or to the proposed CRF.
>
> We provide a fairer comparison between our work and CRFasRNN, as discussed with reviewer 1.

---

### Official Review · AnonReviewer2 · 2018-11-02
**Nice speedup over DenseCRF**

**Rating:** 7
**Confidence:** 4

**Review:**

+ well written
+ Good idea
- Technical section not fully clear
- Some experimental issues

The paper is well written, and clearly explains the background material and concepts. It might almost be a bit too detailed, as the main technical section (4) feels a bit rushed. (more below).

From what I can judge the main idea in the paper is sound. The authors replace the large filtering step in the permutohedral lattice with a spatially varying convolutional kernel. They show that inference is more efficient and training is easier.

The technical section is not very clear. For example: Are the filter weights recomputed for each spatial location, is there any acceleration that speeds this up? How large can the authors make the filter kernel, before the perhutohedral lattice is faster again?

Finally, the experimental section has some room for improvement. I liked the comparison of decoupled and coupled CRF training, but I didn't get much out of the synthetic experiments. I found it particularly confusing since Table 1 doesn't mention that the experiments use ground truth (test) labels that were corrupted.
Second, it would be nice to have a side-to-side comparison between ConvCRF and CRFasRNN. I'd recommend the authors to either use the CRFasRNN training setup for both methods, or spend the week or two training CRFasRNN using their training procedure. It is fine to do either of the two experiments and have four entries in that table.

---

> ### Author Response · Authors · 2018-11-24
> **RE: Nice speedup over DenseCRF**
>
> >> From what I can judge the main idea in the paper is sound. The authors replace the large filtering step in the permutohedral lattice with a spatially varying convolutional kernel. They show that inference is more efficient and training is easier.
>
> This is exactly what the paper is about.
>
> >> Are the filter weights recomputed for each spatial location, is there any acceleration that speeds this up?
>
> Yes, the weights of the filter matrix $k_g$ need to be computed for each spatial location, since they are based in features unique to each location. We implement this by representing $k_g$ as one large 5-dimensional tensor ($k_g[b,dx,dy,x,y]$). For a fixed $dx$ and $dy$ the partial tensor $k_g[:,dx, dy,:,:]$ can be computed using matrix operations on the feature vector $f_i$. Those matrix operations follow the single instruction, multiple data (SIMD) shema and thus can be computed highly efficiently on GPUs (using any common deep learning framework). We don't apply any additional optimization.
>
> >> How large can the authors make the filter kernel, before the perhutohedral lattice is faster again?
>
> In section 5, Table 1 we report the speed of ConvCRF with different filter sizes. The computational time increases quadratically w.r.t. the filter size. So we can estimate that with a filter size of about $50$ both filtering methods have the same speed. Unfortunately we can't experimentally verify this, since our GPU memory can only handle filters up to $31$. This is however not a huge drawback, since we don't observe any improvement in performance for filter-sizes larger than $11$.
>
>
> >> I liked the comparison of decoupled and coupled CRF training, but I didn't get much out of the synthetic experiments. I found it particularly confusing since Table 1 doesn't mention that the experiments use ground truth (test) labels that were corrupted.
>
> The experimental setup for table 1 is discussed in section 5.1. However we have also augmented the caption of the table 1 to make this point clearer (all changes will be uploaded till Monday). The synthetic benchmark serves multiple purposes. Label completion and denoising are tasks in which CRFs have been traditionally strong in. Thus we provide a sound 'hello world' task which can be solved by both CRFs without any training. This also helps us understand the structural differences between the models since optimization is not anymore part of the equation.
>
> >> I'd recommend the authors to either use the CRFasRNN training setup for both methods, or spend the week or two training CRFasRNN using their training procedure. It is fine to do either of the two experiments and have four entries in that table.
>
> This is indeed a good idea. The experiments are currently running and will be finished till Monday. We are going to amend section 5.3 with those results.

---

### Meta-Review · Area_Chair1 · 2018-12-16
**Area chair recommendation**

**Confidence:** 5
**Recommendation:** Reject

**Metareview:**

The authors replace the large filtering step in the permutohedral lattice with a spatially varying convolutional kernel. They show that inference is more efficient and training is easier.

In practice, the synthetic experiments seem to show a greater improvement than appears in real data.  There are concerns about the clarity, lack of theoretical proofs, and at times overstated claims that do not have sufficient support.

The ratings before the rebuttal and discussion were 7-4-6.  After, R1 adjusted their score from 6 to 4.  R2 initially gave a 7 but later said "I think the authors missed an opportunity here. I rated it as an accept, because I saw what it could have been after a good revision. The core idea is good, but fully agree with R1 and R3 that the paper needs work (which the authors were not willing to do). I checked the latest revision (as of Monday morning). None of R3's writing/claims issues are fixed, neither were my additional experimental requests, not even R1's typos." There is therefore a consensus among reviewers for reject.

---

> ### Author Response · Authors · 2018-12-21
> **About R1 comment**
>
> I am surprised that R1 found that nothing was fixed. He might habe opened the wrong PDF. I have done (and included) the additional experiments as requested, addressed R3s issues and fixed the typos of R1. I have uploaded the amended PDF before midnight of the (extended) rebuttal deadline.